# A crystal structure of a collaborative RNA regulatory complex reveals mechanisms to refine target specificity

Chen Qiu[1†], Vandita D Bhat[2†], Sanjana Rajeev[2], Chi Zhang[2], Alexa E Lasley[2], Robert N Wine[1], Zachary T Campbell[2]*, Traci M Tanaka Hall[1]*

[1]Epigenetics and Stem Cell Biology Laboratory, National Institute of Environmental Health Sciences, National Institutes of Health, Research Triangle Park, United States; [2]Department of Biological Sciences, University of Texas at Dallas, Richardson, United States

*For correspondence:
zachary.campbell@utdallas.edu (ZTC);
hall4@niehs.nih.gov (TMTH)

†These authors contributed equally to this work

Competing interests: The authors declare that no competing interests exist.

**Abstract** In the *Caenorhabditis elegans* germline, *fem-3* Binding Factor (FBF) partners with LST-1 to maintain stem cells. A crystal structure of an FBF-2/LST-1/RNA complex revealed that FBF-2 recognizes a short RNA motif different from the characteristic 9-nt FBF binding element, and compact motif recognition coincided with curvature changes in the FBF-2 scaffold. Previously, we engineered FBF-2 to favor recognition of shorter RNA motifs without curvature change (Bhat et al., 2019). In vitro selection of RNAs bound by FBF-2 suggested sequence specificity in the central region of the compact element. This bias, reflected in the crystal structure, was validated in RNA-binding assays. FBF-2 has the intrinsic ability to bind to this shorter motif. LST-1 weakens FBF-2 binding affinity for short and long motifs, which may increase target selectivity. Our findings highlight the role of FBF scaffold flexibility in RNA recognition and suggest a new mechanism by which protein partners refine target site selection.
DOI: https://doi.org/10.7554/eLife.48968.001

## Introduction

RNA-binding proteins control mRNA function. Precise timing of RNA expression, localization, translation and decay permeates virtually every aspect of biology, including pain, memory, and early development (*Bédécarrats et al., 2018*; *Brinegar and Cooper, 2016*; *Conlon and Manley, 2017*; *de la Peña and Campbell, 2018*; *Kershner et al., 2013*; *Nussbacher et al., 2019*; *Shukla and Parker, 2016*). RNA-binding proteins recognize discrete structures and sequences present in untranslated regions (UTRs) (*Mayya and Duchaine, 2019*). They rarely act in isolation. Regulation often requires multiple factors that physically interact. Combinatorial control by multi-protein complexes provides a potential means to diversify regulatory outcomes and to modulate RNA-binding preferences (*Campbell et al., 2012b*; *Hennig et al., 2014*; *Piqué et al., 2008*; *Weidmann et al., 2016*). To interpret mRNA-binding events in cells, understanding how such complexes preferentially recognize their RNA targets is critical.

PUF proteins (named for *Drosophila melanogaster* P̲umilio and *Caenorhabditis elegans* f̲em-3 Binding Factor) are conserved throughout eukaryotes and support a range of processes including development and neurologic function (*Goldstrohm et al., 2018*; *Wickens et al., 2002*). The RNA-binding domain (termed the PUM homology domain) consists of eight α-helical repeats that form a crescent (*Edwards et al., 2001*; *Jenkins et al., 2009*; *Miller et al., 2008*; *Qiu et al., 2012*; *Wang et al., 2002*; *Wang et al., 2001*; *Wang et al., 2009*; *Weidmann et al., 2016*; *Wilinski et al., 2015*; *Zhu et al., 2009*). Along the concave face, RNA is bound in a modular fashion. The 5′ end of the target sequence typically contains a UGU trinucleotide (*Ahringer and Kimble, 1991*;

*Campbell et al., 2012a*; *Dominguez et al., 2018*; *Galgano et al., 2008*; *Gerber et al., 2004*; *Gerber et al., 2006*; *Hafner et al., 2010*; *Morris et al., 2008*; *Wharton and Struhl, 1991*; *White et al., 2001*; *Zamore et al., 1997*; *Zhang et al., 1997*). As PUF proteins lack detectable enzymatic activity, they require partners to assert their regulatory functions. Partners often bind the convex surface of the protein (*Campbell et al., 2012b*; *Edwards et al., 2003*; *Menichelli et al., 2013*; *Weidmann et al., 2016*; *Wu et al., 2013*) or intrinsically-disordered N-terminal regions (*Weidmann and Goldstrohm, 2012*). PUF proteins are largely, but not exclusively, repressive (*Goldstrohm et al., 2018*; *Quenault et al., 2011*). For example, they form complexes with enzymes that promote mRNA decay (*e.g.* the deadenylase CCR4·NOT) and translational repression (*e.g.* Argonaute). PUFs also interact with partners that facilitate mRNA localization (e.g. Myosin) and cytoplasmic polyadenylation (*e.g.* the *C. elegans* GLD-2/GLD-3 poly(A) polymerase complex) (*Friend et al., 2012*; *Goldstrohm et al., 2007*; *Lee et al., 2010*; *Raisch et al., 2016*; *Shen et al., 2009*; *Takizawa and Vale, 2000*; *Van Etten et al., 2012*; *Webster et al., 2019*; *Wu et al., 2013*).

In *C. elegans*, germline stem cells at the distal end of the gonad are maintained by an intricate regulatory network controlled by Notch signaling (*Kershner et al., 2013*). Two transcriptional targets of Notch are LST-1 (Lateral Signaling Target 1) and SYGL-1 (S̲ynthetic G̲erm̲line proliferation defective-1) (*Kershner et al., 2014*). They act redundantly and are required for germline stem cell (GSC) maintenance. Intriguingly, both physically interact in vivo with the homologous and functionally-redundant PUF proteins, FBF-1 and FBF-2 (collectively referred to as FBF). FBF is similarly required for renewal of GSCs (*Crittenden et al., 2002*; *Shin et al., 2017*; *Zhang et al., 1997*). Mechanistically, a key function of FBF is repression of *gld-1* mRNA (*Crittenden et al., 2002*), and LST-1 and SYGL-1 are required for FBF-dependent *gld-1* mRNA repression in the distal germline (*Brenner and Schedl, 2016*; *Shin et al., 2017*). The *gld-1* mRNA contains a 9-nt FBF binding element (FBE) in its 3′UTR that is recognized specifically by FBF (*Crittenden et al., 2002*; *Merritt et al., 2008*). FBF binds more weakly to 8-nt RNA elements that are typically bound by other PUF proteins, such as *C. elegans* PUF-8 (*Opperman et al., 2005*; *Wang et al., 2009*). We recently generated variants of RNA-binding residues in FBF-2 repeat 5 that switch preferential binding of FBF-2 to these 8-nt motifs (*Bhat et al., 2019*).

The molecular details regarding associations between FBF and any of its partner proteins are unknown. Here we investigated the interaction between FBF-2 and LST-1 and the influence of protein partnership on RNA-binding activity. We identified a minimal fragment of LST-1 that forms a tight complex with FBF-2. We determined a crystal structure of the FBF-2/LST-1 complex assembled on an RNA containing a 7mer core sequence that had been identified previously in ~30% of FBF target mRNAs (*Prasad et al., 2016*). The crystal structure revealed a remarkable change in the curvature of FBF-2 that enabled binding to the more compact sequence motif with increased association to RNA bases in the central region of the binding element. This is a new mechanism by which FBF-2 may bind to a shorter RNA element, different from the variants we identified previously that did not change curvature (*Bhat et al., 2019*). In vitro selection and high-throughput sequencing experiments revealed distinct recognition motifs for FBF-2 alone and the FBF-2/LST-1 complex. We confirmed through additional biochemical probing that FBF-2 alone bound the shorter RNA elements that match favored nucleotides at central positions and that LST-1 decreased binding affinity of FBF-2 for both compact and extended RNA elements. We propose a model wherein FBF binds to extended sequence motifs like that found in *gld-1,* where central nucleotides are flipped away from the RNA-binding surface (*Wang et al., 2009*) and also to compact sequence motifs, where it engages the central nucleotides by increasing the curvature of the RNA-binding surface. We propose that partners like LST-1 may restrict the repertoire of targets engaged by FBF in the distal region of the gonad by elevating the importance of affinity. Partners can increase the importance of sequence specificity found in the central region of the binding element.

## Results

### Identification of an LST-1 peptide that is sufficient for interaction with FBF-2

We identified amino acid residues in LST-1 that comprise an interface between LST-1 and FBF-2 using the yeast two-hybrid system (*Figure 1A*). Truncations in LST-1 that encompass residues 1–34 or 34–328 were fused to the LexA DNA-binding domain (*Figure 1—figure supplement 1*) and assayed for their ability to bind FBF-2 fused to the GAL4 activation domain. We found that only the LST-1 34–328 fragment interacted with FBF-2. To narrow the site of interaction, LST-1 was further divided into two regions, residues 34–180 and 180–328. The LST-1 fragment containing residues 34–180 interacted with FBF-2 but residues 180–328 did not interact. Additional truncation of this region revealed that LST-1 residues 34–80 and 80–180 both interacted with FBF-2, as did a minimal fragment containing residues 55–105. A fragment of equivalent length, residues 130–180, did not interact with FBF-2. To examine the specificity of LST-1 interactions, we examined binding to other PUF proteins. As would be predicted, LST-1 interacted with FBF-1 and FBF-2, but did not interact with *C. elegans* PUF-8, *D. melanogaster* Pumilio, human Pum1, or human Pum2 (*Figure 1—figure supplement 2*). We also exchanged the orientation of LST-1 (55–105) and FBF-2 in the two-hybrid experiment by fusing the LST-1 fragment to the GAL4 activation domain and FBF-2 to the LexA DNA-binding domain. We find that either arrangement results in robust interaction between the protein partners (*Figure 1B*). We therefore conclude that residues 55–105 are a minimal interacting fragment of LST-1. This site was also identified independently along with a second weaker interacting site at residues 32–35 (*Haupt et al., 2019*). We inadvertently disrupted the site at residues 32–35 when we removed residues 1–34 in our truncations. We note that our LST-1 fragment containing residues 34–80 has the potential to interact with FBF-2 via residues 34 and 35. Both sites are important for LST-1 activity in vivo, and either site is sufficient for germline stem cell maintenance (*Haupt et al., 2019*).

To identify key amino acid residues for FBF-2 binding, we generated a series of alanine replacements in LST-1 residues 55–105. Studies of association between FBF-2 and protein partners GLD-3 and CPB-1 (Cytoplasmic Polyadenylation element Binding-1) demonstrated that leucine, arginine, and lysine side chains are critical for interaction (*Campbell et al., 2012b*; *Menichelli et al., 2013*; *Wu et al., 2013*). Therefore, we specifically targeted these types of amino acid residues for alanine replacement throughout the 50-aa region spanning residues 55–105. Only the alanine substitution at L83 had a measurable effect in the yeast two-hybrid assay, indicating that L83 was critical for interaction with FBF-2 (*Figure 1B*).

### LST-1 contacts the non-RNA-binding surface of FBF-2 via conserved interaction hot spots

To provide the molecular details of the interaction between FBF-2 and LST-1, we determined a crystal structure of a ternary complex containing the FBF-2 PUM domain, an LST-1 peptide, and RNA. We began by co-expressing the FBF-2 PUM domain and residues 55–105 of LST-1. However, the LST-1 peptide suffered partial degradation after purification of the binary complex from *E. coli* extract. We analyzed the LST-1 peptide that remained associated with FBF-2 by mass spectrometry and identified a shorter fragment of LST-1 (residues 74–98) that formed a complex with FBF-2. We co-expressed FBF-2 with LST-1 residues 74–98 and purified this complex for crystallization. We formed complexes of FBF-2/LST-1 with several different RNAs for crystallization screening. No crystals were obtained of a ternary complex using RNAs containing a 9-nt *gld-1* FBE RNA, 5′-UGUGC-CAUA-3′, with or without an upstream cytosine (*Qiu et al., 2012*). We successfully crystallized a ternary complex of FBF-2 with LST-1 residues 74–98 and an 8-nt RNA that was a highly ranked sequence from in vitro selection experiments (see below) and identified previously in FBF target mRNAs (*Prasad et al., 2016*), 5′-C<u>UGU</u>GAAU-3′. The conserved UGU within this RNA is underlined. By convention, we number the first U of the UGU motif as position +1 and the upstream C as −1. The crystals diffracted to 2.1 Å resolution, and the structure was determined by molecular replacement using the FBF-2 structure (PDB ID: 3V74) as the search model (*Table 1*). Each asymmetric unit contains two ternary complexes. Electron density was visible for LST-1 residues 76–90 for complex A and residues 75–90 for complex B. Complex A includes all eight RNA nucleotides whereas complex

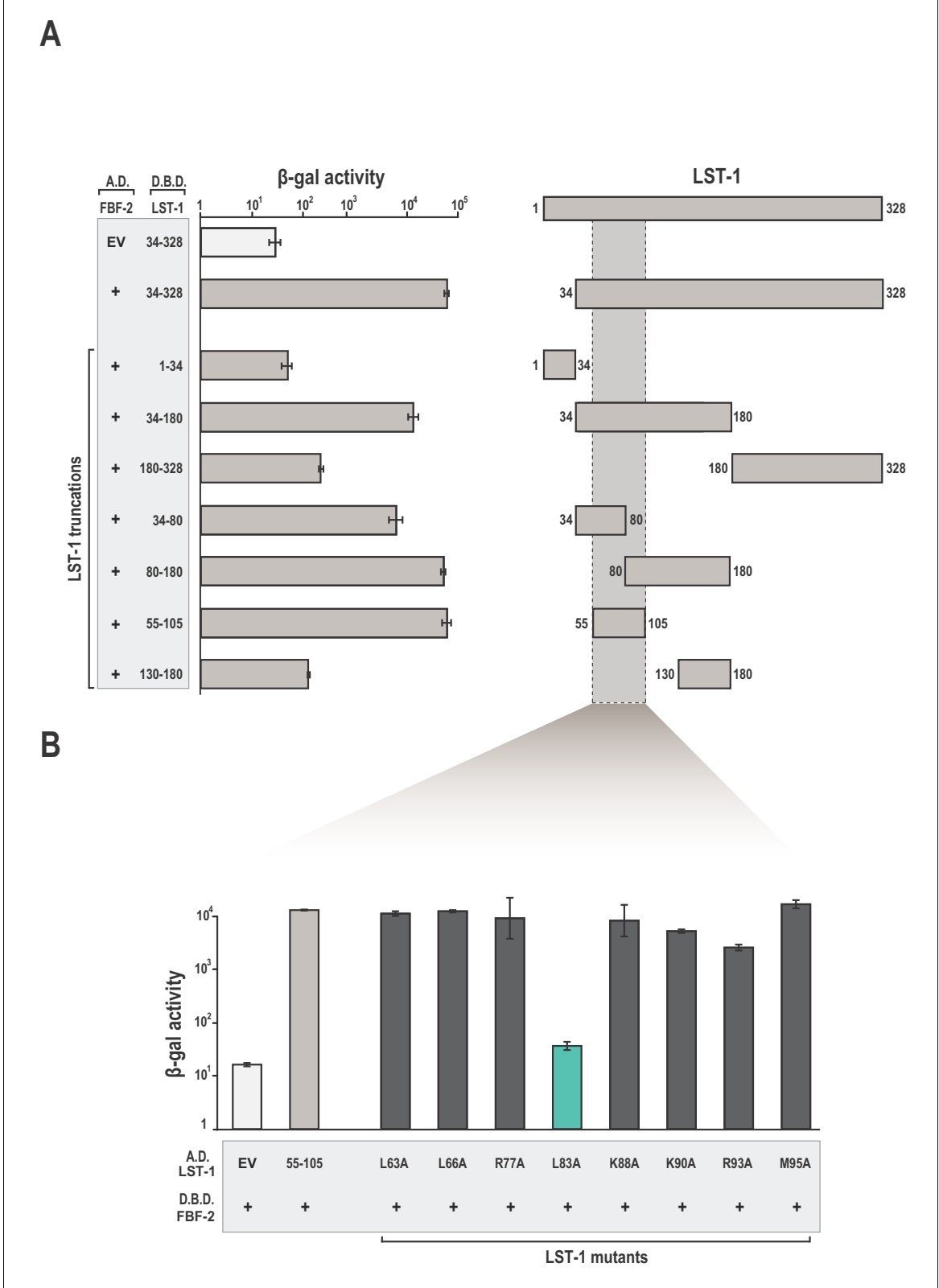

**Figure 1.** Identification of a minimal fragment of LST-1 that interacts with FBF-2. (**A**) Yeast 2-hybrid analyses of interaction between the FBF-2 PUM domain fused to a GAL4 activation domain (A.D.) and LST-1 fragments fused to the LexA DNA-binding domain (D.B.D.). A negative control empty vector (EV) with no FBF-2 fused to the activation domain and a positive control with the FBF-2 PUM domain fused to the activation domain were assessed with LST-1 34–328 fused to the DNA-binding domain and are shown at the top of the graph. (**B**) LST-1 L83 is critical for interaction with FBF-2.
*Figure 1 continued on next page*

*Figure 1 continued*

Yeast 2-hybrid analyses were conducted with LST-1 residues 55–105 fused to a GAL4 activation domain and the PUM domain of FBF-2 fused to the LexA DNA-binding domain. Mutants in LST-1 that interfered with FBF-2 interaction are colored green and those that were competent for interaction are colored gray. Binding activity is shown as units of β-galactosidase (β-gal) activity normalized to cell count. Error bars indicate the standard deviation of three biological replicate measurements. A schematic representation of the yeast 2-hybrid assay is illustrated in *Figure 1—figure supplement 1* and results of yeast 2-hybrid analyses of LST-1 and FBF homologs are shown in *Figure 1—figure supplement 2*.

DOI: https://doi.org/10.7554/eLife.48968.002

The following source data and figure supplements are available for figure 1:

**Source data 1.** Source data for *Figure 1A*-Yeast two-hybrid of WT FBF-2 (A.D.) and LST-1 truncations (D.B.D.).
DOI: https://doi.org/10.7554/eLife.48968.005
**Source data 2.** Source data for *Figure 1B*-Yeast two-hybrid of LST-1 point mutants (A.D.) and WT FBF-2 (D.B.D.).
DOI: https://doi.org/10.7554/eLife.48968.006
**Figure supplement 1.** A schematic of the yeast two-hybrid assay.
DOI: https://doi.org/10.7554/eLife.48968.003
**Figure supplement 2.** LST-1 interacts with FBF but not homologous PUF proteins.
DOI: https://doi.org/10.7554/eLife.48968.004
**Figure supplement 2—source data 1.** Source data for *Figure 1—figure supplement 2*-Yeast two-hybrid of PUF protein homologs (A.D.) and WT LST-1 (D.B.D.).
DOI: https://doi.org/10.7554/eLife.48968.007

B lacks the −1C due to weak electron density. The structures of the two complexes are highly similar: the root mean square deviation (RMSD) is 0.79 Å over 2833 atoms. Below we describe the protein-protein and protein-RNA interactions in complex A.

The crystal structure of the FBF-2/LST-1/RNA ternary complex reveals that LST-1 wraps around the non-RNA-binding surface of FBF-2, making extensive contacts with the C-terminal PUM repeats (*Figure 2A*, *Figure 2—figure supplement 1*). The LST-1 peptide adopts an extended coil conformation and buries a surface area of 839 Å$^2$. We identified three interaction hotspots between FBF-2 and LST-1 (*Figure 2B*): (1) an extended loop between repeats 7 and 8 of FBF-2 (R7-R8 loop) interacts with LST-1 L83, (2) residues in helix α1 of FBF-2 repeat seven interact with LST-1 K80 and the main chain atoms of LST-1 L81, and (3) a hydrophobic pocket between repeats R8 and R8′ of FBF-2 interacts with LST-1 L76. It appears that similar FBF-interacting motifs are present in other partner proteins: LST-1 residues L83, K80, and L76 are conserved in a motif found in the N-terminal region of CPB-1, and L83 and K80 are conserved in a motif found in the C-terminal region of GLD-3, but L76 is substituted with a glutamine in GLD-3 (*Figure 2C*).

To probe the importance of these interaction hotspots, we tested the effects of single residue changes at each of these three hotspots and found that hotspots 1 and 2 are important for FBF-2/LST-1 interaction. Hotspot 1 is critical for FBF-2/LST-1 interaction; as shown above, LST-1 L83A failed to interact with FBF-2 (*Figures 1B* and *2D*). At hotspot 2, LST-1 K80 interacts with S445 and E449 of FBF-2 and FBF-2 Q448 forms hydrogen bonds with main chain atoms of LST-1 L81. We tested the effect of LST-1 K80A or FBF-2 Q448G on FBF-2 and LST-1 interaction. Substitution of K80 with alanine had little to no impact on binding, indicating that this interaction at hotspot 2 is not critical for binding (*Figure 2D*). Weak electron density for the K80 side chain suggests that this interaction is not ordered in the crystal structure (*Figure 2—figure supplement 1*). FBF-2 Q448G had a minor effect on LST-1 interaction (*Figure 2E*), suggesting that the interaction of Q448 with the LST-1 backbone at hotspot 2 contributes to binding affinity (*Figure 2B*, *Figure 2—figure supplement 1*). In contrast to hotspots 1 and 2, the third hotspot at L76 was largely dispensable for binding, which may explain its lack of conservation in GLD-3 (*Figure 2C*). We conclude that interactions with LST-1 residues C-terminal to K80 are critical for binding, which is consistent with the strength of binding of the deletion construct containing LST-1 residues 80–180.

The FBF-2/LST-1/RNA crystal structure indicates that LST-1 L83 interacts with FBF-2 at the base of the R7-R8 loop, which had previously been identified as the site of interaction between FBF-2 and binding partners CPB-1 or GLD-3 (*Figures 2C* and *3A*) (*Campbell et al., 2012b*; *Menichelli et al., 2013*; *Wu et al., 2013*). Probing this interaction by mutagenesis identifies similarities and differences in FBF-2 interaction with these three proteins. In crystal structures of FBF-2 binary complexes with RNA, the FBF-2 R7-R8 loop was disordered, but in our structures of the FBF-2/LST-1/RNA ternary

**Table 1.** X-ray data collection and refinement statistics.

| Resolution range | 39.7–2.1 (2.174–2.1) |
|---|---|
| Space group | P 1 |
| Unit cell dimensions a, b, c (Å) <br> α, β, γ (°) | 42.75, 74.38, 81.55 <br> 107.17, 104.40, 101.76 |
| Total reflections[*] | 180,242 (13587) |
| Unique reflections | 26,619 (4934) |
| Multiplicity | 6.8 (7.0) |
| Completeness (%) | 96.6 (95.3) |
| Mean I/sigma(I) | 11.8 (2.5) |
| Wilson B-factor | 41.2 |
| R-merge | 0.101 (0.795) |
| R-meas | 0.109 (0.858) |
| R-pim | 0.041 (0.322) |
| CC$_{1/2}$ | 0.995 (0.885) |
| Refinement | |
| Reflections used in refinement | 50,102 (4931) |
| Reflections used for R-free | 2000 (197) |
| R-work | 0.198 (0.296) |
| R-free | 0.240 (0.343) |
| Number of atoms | |
| protein | 6565 |
| RNA | 266 |
| Solvent | 189 |
| RMSD bonds (Å) | 0.003 |
| RMSD angles (°) | 0.82 |
| Ramachandran favored (%) | 98.38 |
| Ramachandran allowed (%) | 1.62 |
| Ramachandran outliers (%) | 0.00 |
| Average B-factors (Å$^2$) | |
| protein | 53.6 |
| RNA | 76.7 |
| solvent | 52.2 |

[*]Statistics for the highest-resolution shell are shown in parentheses.

DOI: https://doi.org/10.7554/eLife.48968.008

complex, the loop is visible and forms a pocket for interaction with LST-1 L83 (*Figure 3A*). Because L83 is critical for interaction with FBF-2, we also probed the importance of the FBF-2 R7-R8 loop. Deleting residues Y479-T485 of FBF-2 abrogated binding to LST-1 (*Figure 3B*). We next interrogated the importance of individual residues of the loop by alanine scanning mutagenesis and found that only a Y479A mutation disrupted LST-1 interaction (*Figure 3B*). FBF-2 Y479 is at the base of the R7-R8 loop and forms part of a hydrophobic binding pocket for LST-1 L83 (*Figure 3A*). We tested the effects of other amino acid substitutions for Y479. Mutations to glycine, glutamine, valine, phenylalanine, or arginine also disrupted FBF-2 interaction with LST-1 (*Figure 3C*), indicating the central importance of Y479. This result is highly reminiscent of prior findings with CPB-1 and GLD-3, which are highly dependent on Y479 (*Campbell et al., 2012b*; *Menichelli et al., 2013*; *Wu et al., 2013*). LST-1 Y85 binds near FBF-2 Y479, and a Y85A mutation disrupted FBF-2/LST-1 interaction, supporting the importance of Y479 (*Figure 2D*). FBF-2 L444A, I480A, and T485A mutations did not affect LST-1 interaction (*Figure 3B*), despite their importance for interaction with CPB-1 (*Campbell et al.,*

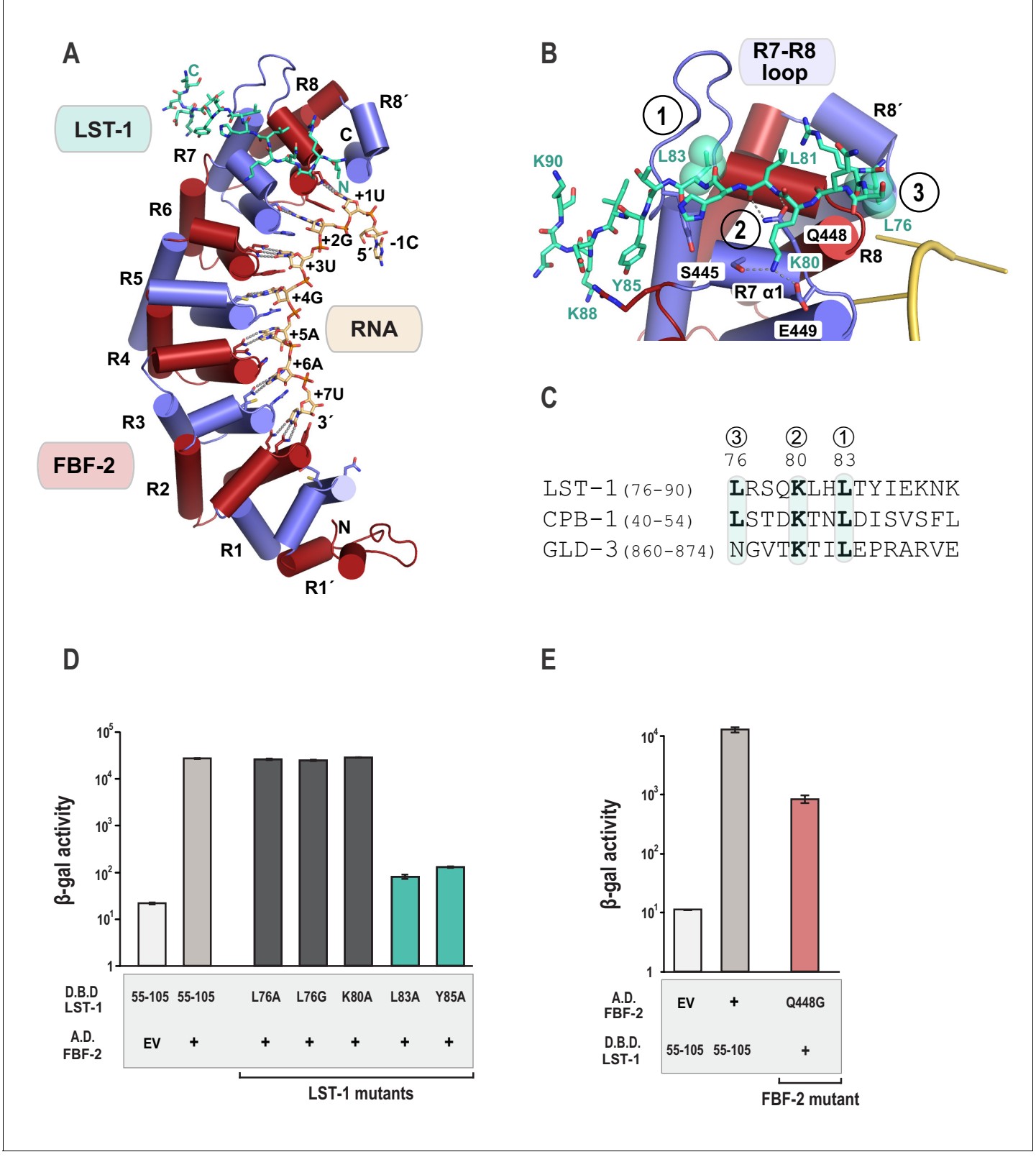

**Figure 2.** Crystal structure of an FBF-2/LST-1/RNA ternary complex reveals hotspots for protein-protein interaction. (**A**) Crystal structure of an FBF-2/LST-1/RNA ternary complex. FBF-2 is shown as a ribbon diagram with cylindrical helices. PUM repeats are colored alternately red and blue. RNA recognition side chains from each PUM repeat are shown with dotted lines indicating interactions with the RNA bases. LST-1 (green) and the RNA (beige) are shown as stick representations colored by atom type (red, oxygen; blue, nitrogen; orange, phosphorus). (**B**) LST-1 contacts FBF-2 at

*Figure 2 continued on next page*

Figure 2 continued

conserved interaction hotspots. Zoomed-in view of interaction between FBF-2 and LST-1. Three interaction hotspots are labeled, and LST-1 L83 and L76 at hotspots 1 and 3, respectively, are shown with space-filling atoms. LST-1 K80 and FBF-2 Q448 at hotspot 2 are shown as stick models. Interactions between LST-1 and FBF-2 are indicated by dotted lines. Electron density for the LST-1 peptide is shown in *Figure 2—figure supplement 1*. (C) Conservation of LST-1 interacting residues in CPB-1 and GLD-3. Amino acid sequence alignment of the LST-1 interacting peptide and conserved sequences in CPB-1 and GLD-3. Residues at the interaction hotspots in (B) are highlighted and conserved residues are in boldface. (D) LST-1 L83 and Y85 at interaction hotspot 1 are essential for tight binding to FBF-2. Yeast 2-hybrid analyses were conducted with LST-1 residues 55–105 fused to the LexA DNA-binding domain (D.B.D.) and the PUM domain of FBF-2 fused to the GAL4 activation domain (A.D.). Mutants in LST-1 that interfered with FBF-2 interaction are colored green and those that were competent for interaction are colored gray. (E) FBF-2 Q448G at hotspot 2 has a minor effect on interaction with LST-1. FBF-2 variants that interfered with LST-1 interaction are colored red and those that were competent for interaction are colored gray. Binding activity is shown as units of β-gal activity normalized to cell count. Error bars indicate the standard deviation of three biological replicate measurements.

DOI: https://doi.org/10.7554/eLife.48968.009

The following source data and figure supplement are available for figure 2:

**Source data 1.** Source data for *Figure 2D*-Yeast two-hybrid of LST-1 point mutants (D.B.D.) and WT FBF-2 (A.D.).
DOI: https://doi.org/10.7554/eLife.48968.011
**Source data 2.** Source data for *Figure 2E*-Yeast two-hybrid of FBF-2 point mutants (A.D.) and WT LST-1 (D.B.D.).
DOI: https://doi.org/10.7554/eLife.48968.012
**Figure supplement 1.** $F_o$-$F_c$ simulated annealing omit map for the LST-1 peptide, contoured at 3 σ.
DOI: https://doi.org/10.7554/eLife.48968.010

*2012b*). L444 is adjacent to Y479 and also forms part of the L83 binding pocket (*Figure 3A*). I480 and T485 are within the R7-R8 loop but their side chains do not interact with LST-1 L83. The amino acid sequences of LST-1, CPB-1, and GLD-3 vary around the conserved leucine residue (L83 in LST-1, *Figure 2C*), which may shift the relative importance of FBF-2 residues for interaction with each protein.

## The crystal structure of the ternary complex reveals an altered RNA-binding mode and interaction surface curvature of FBF-2

The crystal structure of the FBF-2/LST-1/RNA ternary complex revealed two striking changes in the FBF-2/RNA interaction versus what was observed previously in crystal structures of FBF-2 alone bound to FBE RNAs: (1) the central PUM repeats 4 and 5 of FBF-2 bind to the RNA in a 1-repeat-to-1-nucleotide pattern and (2) the FBF-2 protein curvature is more pronounced in the ternary complex. In the 1:1 interaction mode, base-interacting residues of repeat 4 recognize A5 and those of repeat 5 recognize G4, including stacking of R364 between nucleotides G4 and A5 (*Figure 4A*, *Figure 4—figure supplement 1*). In contrast, FBF-2 alone binds to *gld-1* RNA with triply-stacked RNA bases 4–6 flipped away from the RNA-binding surface, and FBF-2 repeat 4 does not contact the RNA (*Figure 4B*). When we compared the overall FBF-2 structure in the ternary complex with that in a binary complex of FBF-2, we found that the RNA-binding surface of the FBF-2 protein in the ternary complex is more curved than that in the binary complex (*Figure 4C*). Since the curvature of PUF proteins often correlates with RNA-binding motif length specificity, the increased FBF-2 RNA-binding surface curvature in the ternary complex appears to support transition to the 1:1 RNA recognition mode.

The shift to a 1:1 RNA recognition mode results in FBF-2 recognizing a more 'compact' motif than the 'extended' 9-nt motif we had observed previously for FBF-2 alone. The RNA in the FBF-2/LST-1/RNA ternary complex is two nucleotides shorter than in binary complexes. This is due to loss of the directly stacked nucleotides 4–6 and also because repeat 1 is not engaged in RNA binding. Although we previously had engineered FBF-2 to recognize a compact 8-nt motif using a 1:1 RNA recognition mode, we had not observed a 1:1 RNA recognition mode for wild-type FBF-2 (*Bhat et al., 2019*). We reasoned that the change in binding mode could be due to LST-1 binding.

## LST-1 imparts distinct RNA-binding selectivity to FBF-2

To determine whether the presence of LST-1 modulated the RNA-binding specificity of FBF-2, we examined sequence preferences using SEQRS (in vitro <u>se</u>lection, high-throughput sequencing of <u>R</u>NA, and <u>s</u>equence specificity landscapes, *Figure 5A*) and found that LST-1 appears to alter the 3′

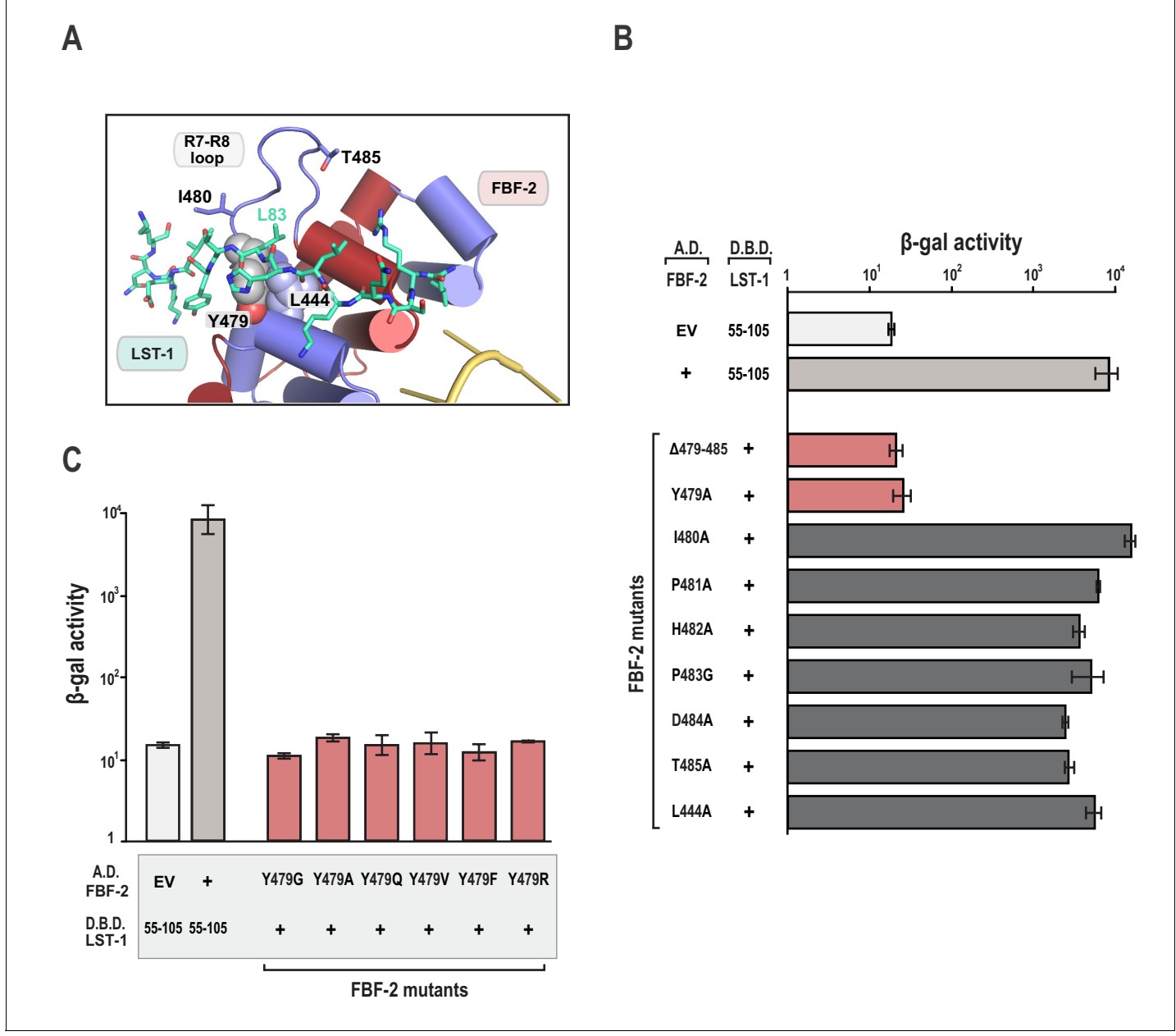

**Figure 3.** The FBF-2 R7-R8 loop is essential for interaction with LST-1. (**A**) The essential residue LST-1 L83 interacts with FBF-2 at the base of the FBF-2 R7-R8 loop. FBF-2 L444 and Y479 at the R7-R8 loop are shown with space-filling atoms. (**B**) Yeast 2-hybrid analyses were conducted with LST-1 residues 55–105 fused to the LexA DNA-binding domain (D.B.D.) and the PUM domain of FBF-2 fused to the GAL4 activation domain (A.D.). (**C**) Yeast 2-hybrid analyses of mutations in Y479. Mutants in FBF-2 that interfered with LST-1 interaction are colored red and those that were competent for interaction are colored gray. Binding activity is shown as units of β-gal activity normalized to cell count. Error bars indicate the standard deviation of three biological replicate measurements.

DOI: https://doi.org/10.7554/eLife.48968.013

The following source data is available for figure 3:

**Source data 1.** Source data for *Figure 3B*-Yeast two-hybrid of FBF-2 point mutants (A.D.) and WT LST-1 (D.B.D.).
DOI: https://doi.org/10.7554/eLife.48968.014
**Source data 2.** Source data for *Figure 3C*-Yeast two-hybrid of FBF-2 point mutants (A.D.) and WT LST-1 (D.B.D.).
DOI: https://doi.org/10.7554/eLife.48968.015

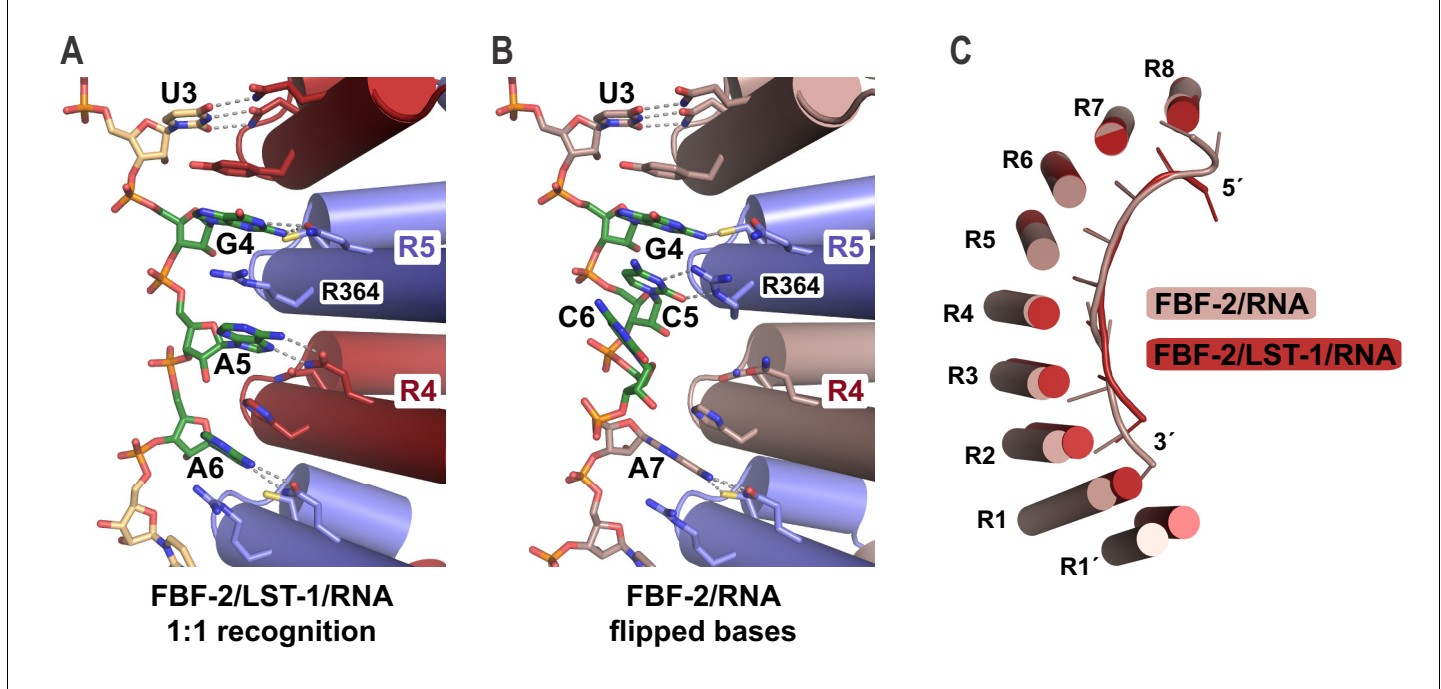

**Figure 4.** FBF-2 in the ternary complex binds to RNA using a 1:1 recognition mode and its curvature is more pronounced. (**A**) FBF-2 recognizes the central nucleotides in a compact RNA using repeats 4 and 5. The crystal structure of the FBF-2/LST-1/RNA ternary complex is shown with FBF-2 displayed as a ribbon diagram with cylindrical helices. PUM repeats are colored alternately red and blue. RNA recognition side chains from each PUM repeat are shown with dotted lines indicating interactions with the RNA bases. Central nucleotides 4–6 (green) within a compact RNA element (beige) are shown as stick representations colored by atom type (red, oxygen; blue, nitrogen; orange, phosphorus). Electron density for the compact RNA nucleotides 4–6 is shown in *Figure 4—figure supplement 1*. (**B**) FBF-2 binds to directly stacked and flipped central nucleotides in the extended *gld-1* RNA motif. The crystal structure of the FBF-2/*gld-1* RNA binary complex (PDB ID 3V74) is shown as a ribbon diagram with cylindrical helices. Central nucleotides 4–6 (green) within the *gld-1* RNA (mauve) are shown as stick models. (**C**) Superposition of FBF-2 within ternary and binary complexes reveals increased curvature in the FBF-2/LST-1/RNA ternary complex. RNA-binding helices and RNA cartoons are shown for FBF-2 in the binary (mauve) and ternary (red) complexes.

DOI: https://doi.org/10.7554/eLife.48968.016

The following figure supplement is available for figure 4:

**Figure supplement 1.** $F_o$-$F_c$ simulated annealing omit map for the cFBE RNA nucleotides 4–6, contoured at 3 σ.

DOI: https://doi.org/10.7554/eLife.48968.017

sequence specificity of FBF-2 (*Campbell et al., 2012a*; *Lou et al., 2017*; *Zhou et al., 2018*). To obtain the specificity of the complex, LST-1 (residues 34–180) was immobilized on glutathione magnetic resin. FBF-2 (residues 163–632) was purified as a fusion to the maltose binding protein. Incubation of FBF-2 with LST-1 enabled affinity capture of FBF-2 and formation of a protein complex. A random RNA library was added to the complex, and unbound RNAs were removed with washing. Bound RNAs were reverse transcribed and amplified by PCR. One of the amplification primers contains the promoter element for T7 RNA polymerase. Thus, the dsDNA product is used as a template for the subsequent round of selection. After five rounds, the sample was subjected to high-throughput sequencing. The motifs of the FBF-2/LST-1 complex and FBF-2 alone displayed a conserved 5′ sequence element (*Figure 5B,C*). As a key negative control, we analyzed the specificity of LST-1 which failed to yield a motif with high information content. The FBF-2/LST-1 motif was reminiscent of the SEQRS motif of the *Drosophila melanogaster* Pum/Nos complex, which loses 3′ sequence selectivity relative to that of Pum alone (*Weidmann et al., 2016*). For both the Pum/Nos and FBF-2/LST-1 complexes, our crystal structures indicate that Pum and FBF-2 recognize the nucleotides at the 3′ end, despite the variability of the sequence in that region.

The Pum/Nos complex binds more tightly to target RNAs than Pum alone, but in contrast, we found that LST-1 weakened the affinity of FBF-2. Moreover, LST-1 was not required to permit FBF-2 binding to the compact RNA element. We measured RNA-binding affinities of FBF-2 and FBF-2/LST-

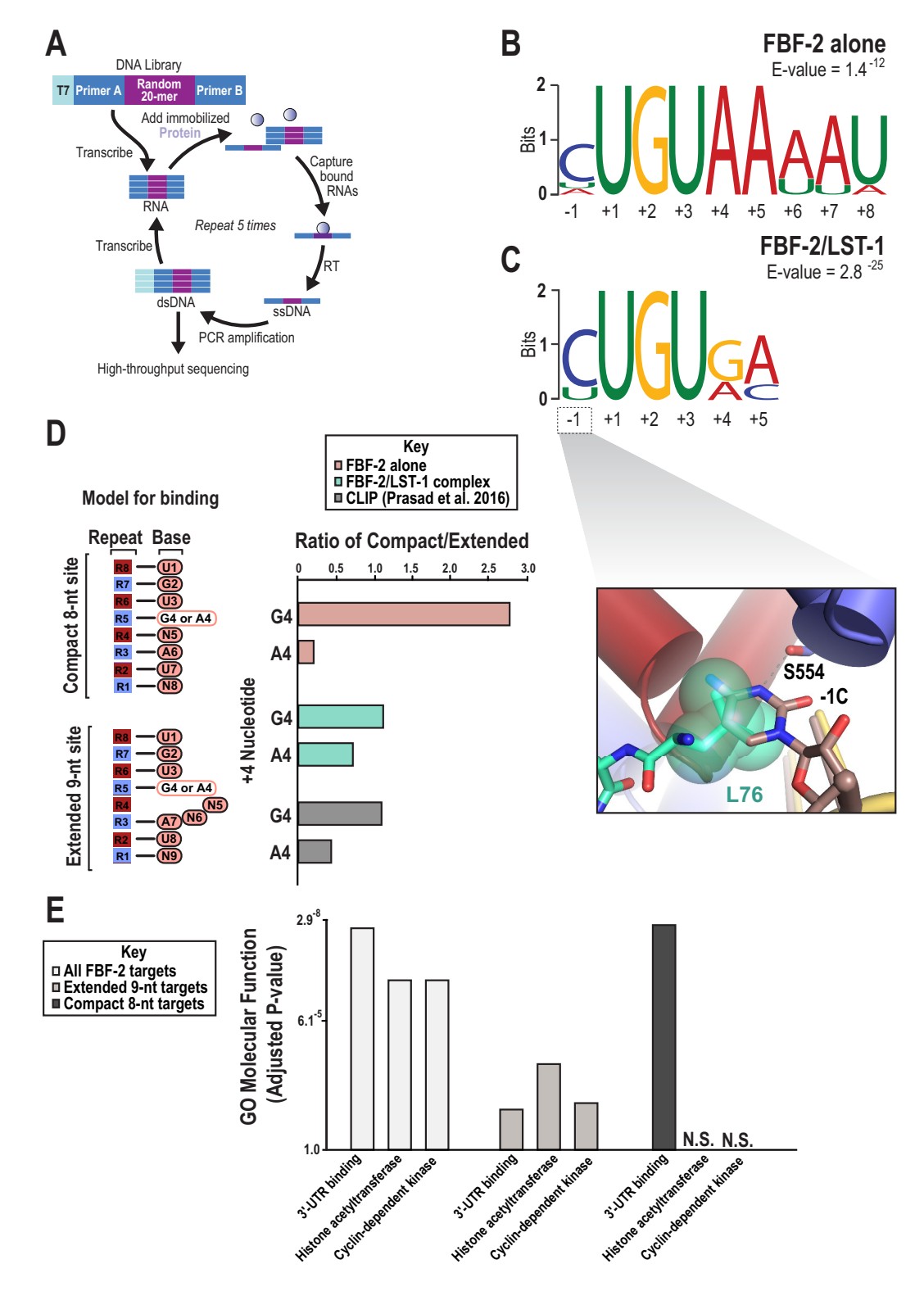

**Figure 5.** SEQRS analysis of FBF-2/LST-1 and FBF-2 reveals distinct specificities. (**A**) Diagram of the SEQRS procedure. (**B**) Motif from SEQRS analysis of the FBF-2/LST-1 complex. (**C**) Motif from SEQRS analysis of FBF-2. Inset, superposition of the upstream C pocket in structures of the FBF-2/LST-1/RNA ternary and FBF-2/RNA binary complexes demonstrates that LST-1 L76 occupies the upstream C pocket in the structure of the ternary complex. (**D**) Comparative analysis of biases at base +4 in compact vs extended motifs. Sequences that conform to either the compact 8-nt or extended 9-nt sites

*Figure 5 continued on next page*

*Figure 5 continued*

were quantified in SEQRS data for FBF-2 alone (pink), the LST-1/FBF-2 complex (cyan), or CLIP data for FBF-2 (gray). (E) GO term analysis of FBF-2 mRNA targets. P-values were corrected using the Benjamini-Hochberg method (*Kuleshov et al., 2016*). Enrichment for compact sequences or extended binding elements was determined using the grep command on FBF-2 CLIP targets (*Prasad et al., 2016*). The abbreviation N.S. indicates that enrichment failed to achieve significance (adjusted p<0.05).

DOI: https://doi.org/10.7554/eLife.48968.018

The following source data and figure supplements are available for figure 5:

**Source data 1.** Source data for *Figure 5B,C*-Sequences for MEMEs.
DOI: https://doi.org/10.7554/eLife.48968.020

**Source data 2.** Source data for *Figure 5E*-mRNA targets for GO term enrichment.
DOI: https://doi.org/10.7554/eLife.48968.021

**Figure supplement 1.** Representative EMSA gels and corresponding binding curves are shown for binding to *gld-1* (A) and compact FBE (cFBE, (B) RNAs.
DOI: https://doi.org/10.7554/eLife.48968.019

**Figure supplement 1—source data 1.** Source data for *Figure 5—figure supplement 1* and *Table 2*-$K_d$ values for triplicate measurements.
DOI: https://doi.org/10.7554/eLife.48968.022

---

1 residues 67–98 by electrophoretic mobility shift assay (EMSA). We performed the EMSAs using the same preparation of FBF-2 and added a constant concentration of 4 µM LST-1 to assure complex formation even at lower FBF-2 concentrations. FBF-2 and LST-1 regulate expression of *gld-1* mRNA in the *C. elegans* germline, so we tested binding of FBF-2 and FBF-2/LST-1 to the extended RNA element in the *gld-1* FBEa sequence (5′-CAUGUGCCAUA-3′). We found that LST-1 weakened binding affinity of FBF-2 for this element 4-fold ($K_d$ 12 nM vs 46 nM) (*Table 2*, *Figure 5—figure supplement 1*). We also measured binding of FBF-2 to the RNA sequence in the crystal structure (cFBE-7 RNA) and found that FBF-2 alone bound tightly to the compact element ($K_d$ 22 nM), and as we observed for the *gld-1* extended element, the FBF-2/LST-1 complex bound more weakly ($K_d$ 112

**Table 2.** RNA-binding analyses of FBF-2 and FBF-2/LST-1[1].

| RNA | 87654 321 rpt<br>C−UGUGA−AUG (8)<br>C−UGUGCCAUA (9)<br>12345 pos[2] | FBF-2,<br>$K_d$ (nM) | $K_{rel}$[2] | FBF-2/LST-1,<br>$K_d$ (nM) | $K_{rel}$[3] |
|---|---|---|---|---|---|
| *gld-1* FBEa | CAUGUGCCAUA | 12.4 ± 2.0 | 1 | 46.4 ± 5.0 | 1 |
| *gld-1* –2U | **U**AUGUGCCAUA | 32.2 ± 4.7 | 2.6 | 101.3 ± 13.2 | 2.2 |
| *gld-1* G4A | CAUGU**A**CCAUA | 12.0 ± 1.4 | 1 | 34.4 ± 5.6 | 0.7 |
| *gld-1* C5A | CAUGUG**A**CAUA | 27.1 ± 5.4 | 2.2 | 79.2 ± 8.8 | 1.7 |
| cFBE-7 | C−UGUGA−AU | 22.0 ± 2.7 | 2.1 | 111.7 ± 7.7 | 2.9 |
| cFBE | C−UGUGA−AUG | 10.3 ± 2.9 | 1 | 38.7 ± 5.0 | 1 |
| cFBE −1U | **U**−UGUGA−AUG | 46.5 ± 4.3 | 4.5 | 175.9 ± 37.8 | 4.5 |
| PBE | C−UGU**AU**−AU**A** | 56.8 ± 13.7 | 5.5 | 814.0 ± 180 | 21 |
| cFBE G4A | C−UGU**A**A−AUG | 18.8 ± 3.0 | 1.8 | 82.7 ± 16.0 | 2.1 |
| cFBE A5C | C−UGUG**C**−AUG | 19.5 ± 2.5 | 1.9 | 82.3 ± 18.8 | 2.1 |
| cFBE A5U | C−UGUG**U**−AUG | 25.5 ± 5.5 | 2.5 | 133.2 ± 23.8 | 3.4 |
| cFBE G8A | C−UGUGA−AU**A** | 21.1 ± 2.5 | 1.9 | 84.4 ± 20.2 | 2.2 |

[1]Representative EMSA gels and binding curves are shown in *Figure 5—figure supplement 1*. Source data for the three technical replicate EMSAs are included in *Figure 5—figure supplement 1—source data 1*.

[2]RNA sequences of the cFBE compact element and *gld-1* FBEa motif are shown with the FBF-2 repeat (rpt) that binds to the respective nucleotide above and the RNA motif position below. Nucleotides in boldface differ from the sequences of the *gld-1* FBEa motif (top four lines) or the cFBE.

[3]Relative $K_d$ values ($K_{rel}$) are calculated with respect to the $K_d$ for binding to the *gld-1* FBEa motif (top four lines) or the cFBE.

DOI: https://doi.org/10.7554/eLife.48968.023

nM) (*Table 2*). We added an additional G at the 3′ end to potentially occupy the repeat 1 binding site (cFBE RNA), and this RNA bound 2-to-3-fold more tightly to both FBF-2 ($K_d$ 10 nM) and FBF-2/LST-1 ($K_d$ 39 nM). We conclude that FBF-2 has the intrinsic ability to bind to extended and compact RNA elements, and LST-1 weakens binding affinity for both types of targets. We therefore explored whether the differences in SEQRS motifs were due to distinct sequence specificities.

## FBF-2 binds to compact elements bearing an upstream C and G4 and A5 central nucleotides

FBF-2 binds specifically to a cytosine at either position −1 or −2 upstream of the UGU trinucleotide (*Qiu et al., 2012*), and this was reflected in the SEQRS analysis of both FBF-2/LST-1 and FBF-2 alone (*Figure 5B,C*). This result was surprising since the RNA in the FBF-2/LST-2/RNA ternary complex contains a −1C and the crystal structure of the complex indicated that the −1C base was not bound in the upstream C binding pocket that is marked by FBF-2 S554. In complex A, the electron density showed clearly that the −1C nucleotide was flipped away from FBF-2 and did not contact the protein (*Figure 2A*). In complex B, the −1C base was disordered with no visible density. Superposition of structures of the FBF-2/LST-1/RNA ternary complex and an FBF-2/RNA binary complex revealed that LST1 L76 at interaction hotspot 3 protrudes into the cytosine binding pocket, which could prevent or weaken the effect of −1C binding (*Figure 5C* inset). This structural feature suggested that LST-1 might alter the upstream C binding preference of FBF-2.

We tested by EMSA whether an upstream C affected binding affinity of FBF-2 in complex with LST-1 and found that FBF-2 binds more tightly to RNAs with an upstream C, even in the presence of LST-1, consistent with the SEQRS motifs. When we changed the upstream C to a U in the *gld-1* RNA, binding affinities were ~2 fold weaker for both FBF-2 and FBF-2/LST-1, indicating that the upstream C binding pocket is still used by the FBF-2/LST-1 complex when interacting with an extended element (*Table 2*, *Figure 5—figure supplement 1*). We also compared binding affinities of FBF-2 and FBF-2/LST-1 for the compact RNA element with either an upstream −1C or −1U. Replacing the upstream C with U in the compact element weakened binding affinity 4.5-fold for both FBF-2 and FBF-2/LST-1. Therefore, FBF-2 binds more tightly to both extended and compact elements with an upstream C.

The SEQRS motifs suggested that the FBF-2/LST-1 complex could differ from FBF-2 alone in sequence specificity at positions +4 and +5, and we found that FBF-2 binding affinity for compact elements is sensitive to substitutions at these positions. We previously showed that FBF-2 binds to an 8-nt PBE with the core sequence 5′-UGUAAAUA-3′, although it prefers an extended *gld-1* motif (*Bhat et al., 2019*). We tested binding of FBF-2 and FBF-2/LST-1 to another 8-nt PBE with the core sequence 5′-UGUAUAUA-3′. FBF-2 bound this RNA with 6-fold weaker affinity than the compact element in the crystal structure, and the FBF-2/LST-1 complex bound it very poorly (*Table 2*). These sequences differ from the new compact motif (5′-UGUGAAUG-3′) at positions +4, +5, and +8. We systematically tested the three individual substitutions (G4A, A5U, and G8A) and found that each substitution weakened binding affinity 2–3-fold. A compact element with an A5C substitution also bound 2-fold weaker. Therefore, FBF-2 favors G4, A5, and 8G in a compact motif. In contrast, FBF-2 bound equally well to RNAs with A4 or G4 in the extended element. An extended *gld-1* element with a C5A substitution bound 2-fold weaker, indicating that FBF-2 has a slight preference for C at position +5 of an extended element (*Table 2*). Although we measured a modest 2-fold stronger affinity for G8 over A8 in our EMSAs, the +8 position in the compact motif and the +9 position in the extended motif are degenerate in the SEQRS analysis (*Figure 5B,C*). Our EMSA data are clear that LST-1 is not required for FBF-2 binding to the compact RNA element and it weakens FBF-2 binding affinity for all of the RNAs we tested. We suggest that weaker binding affinity might make the complex more sensitive to the identity of central nucleotides and therefore favored selection of compact elements that match the preferred G4 and A5 in our SEQRS experiments.

In addition to sequence preferences, we examined whether the identity of the nucleotide at position +4 influenced the FBF-2 sequence motif. Our EMSA results indicated that a compact motif with a G4 bound with higher affinity than a motif with an A4, but substitution at the +4 position had no effect on binding to an extended motif. We asked whether the identity of the +4 nucleotide (A or G) affects the length of motif and found that a G4 is associated with a compact motif and an A4 is associated with an extended motif. We counted occurrences of sequences in the SEQRS data that represent the variations at the +4 position and also sorted the sequences by compact or extended motifs:

extended motifs with A4 (5´-CUGU**A**nnAU, where n = any nucleotide), extended motifs with G4 (5´-CUGU**G**nnAU), compact motifs with A4 (5´-CUGU**A**nAU), and compact motifs with G4 (5´-CUGU**G**-nAU) (*Figure 5D*, *Table 3*). We used the SEQRS data from FBF-2 alone, which contains more sequencing reads than the FBF-2/LST-1 dataset. We found that sequences with an A4 were predominantly extended motifs (1:5 compact:extended motif ratio). In contrast, sequences with a G4, although less frequently selected, were predominantly compact motifs (3:1 compact:extended motif ratio). We had fewer sequencing reads for the FBF-2/LST-1 SEQRS experiment, possibly due to the weaker binding affinity of the complex, making it more difficult to draw strong conclusions from those SEQRS data. We also found evidence that these patterns hold in vivo. By looking at sequence motifs at sites that crosslinked to FBF-2 in worms, which represent activity of FBF-2 alone or with different partner proteins, we found that an A4 was more often in an extended motif and a G4 showed a slight bias toward a compact motif (*Figure 5D*, *Table 3*). Moreover, the compact motif that was identified in 30% of CLIP sites is enriched in G4 and A5 (*Prasad et al., 2016*).

We sought to determine if the compact motifs bound by FBF were found in mRNA targets with biological functions distinct from those bearing extended motifs. We again used the experimentally defined targets of FBF-2 to identify CLIP binding sites containing either the compact (CUGURnAU) or extended (CUGURnnAU) binding elements (*Prasad et al., 2016*). Using gene ontology analysis, we found that all FBF-2 targets are highly enriched for three functions – 3´ UTR RNA binding, histone acetyl transferase activity, and cyclin dependent kinase activity (*Figure 5E*). While all three functions are found in transcripts bound by FBF-2 in vivo, only a subset containing 3´ UTR binding factors were enriched in transcripts with the compact sequence. This implies that binding element length is related to specification of regulatory networks, reminiscent of similar analyses on PUF protein regulatory networks in yeast (*Valley et al., 2012*; *Wilinski et al., 2017*; *Wilinski et al., 2015*).

## Discussion

Our study examines how a specific protein partner expressed in the distal end of the *C. elegans* germline modulates the interaction of a conserved RNA-binding protein with target transcripts. We identified a segment of LST-1 outside of any predicted protein domains, L76-K90, that binds to FBF-2 in cells and in vitro. The crystal structure of the complex revealed that the site of interaction was localized to a region of FBF-2 implicated in binding to multiple protein partners. The structural information suggested four unanticipated characteristics of FBF-2 and LST-1 function: FBF-2 curvature flexibility, specific recognition by FBF-2 of central nucleotides in a distinct motif, recruitment to a

**Table 3.** SEQRS enrichment for specific sequence elements.

| Protein | Pattern | Base +4 | Terminal AU position | Count | Ratio compact/ extended |
|---|---|---|---|---|---|
| | 87654 321 repeat<br>CUGUGA AUG (8mer)<br>CUGUGCCAUA (9mer) | | | | |
| FBF-2 | CTGTA..AT | A | +8U | 119374 | 0.21 |
| FBF-2 | CTGTA. AT | A | +7U | 24819 | |
| FBF-2 | CTGTG..AT | G | +8U | 1970 | 2.8 |
| FBF-2 | CTGTG. AT | G | +7U | 5506 | |
| Complex | CTGTA..AT | A | +8U | 170 | 0.7 |
| Complex | CTGTA. AT | A | +7U | 118 | |
| Complex | CTGTG..AT | G | +8U | 113 | 1.1 |
| Complex | CTGTG. AT | G | +7U | 126 | |
| CLIP | CTGTA..AT | A | +8U | 266 | 0.44 |
| CLIP | CTGTA. AT | A | +7U | 117 | |
| CLIP | CTGTG..AT | G | +8U | 92 | 1.1 |
| CLIP | CTGTG. AT | G | +7U | 102 | |

DOI: https://doi.org/10.7554/eLife.48968.024

position utilized by multiple protein-partners, and weakened binding affinity for the LST-1/FBF-2 complex.

First, the curvature of FBF-2 can be altered to bind to RNA in a 1-repeat-to-1-RNA base mode. All previous crystal structures of FBF-2 in complex with a variety of RNA sequences retained the same curvature upon RNA binding (*Bhat et al., 2019*; *Koh et al., 2011*; *Qiu et al., 2012*; *Wang et al., 2009*). The only crystal structures of PUF-like proteins in apo and RNA-bound forms that have revealed a large conformational change are those of *S. cerevisiae* Nop9 protein. Nop9 is an atypical PUF-like protein with 11 PUM repeats that binds to both structured and single-stranded RNAs, and RNA binding decreases curvature of the protein (*Wang and Ye, 2017*; *Zhang et al., 2016*). In contrast, crystal structures of human Pum1 and *S. cerevisiae* Puf4 proteins alone and in complex with RNA show no alterations in curvature upon RNA binding (*Miller et al., 2008*; *Wang et al., 2002*; *Wang et al., 2001*). On the other hand, differences in the curvature among PUF proteins are well established and are correlated with the length of sequence motif recognized. For example, yeast PUF proteins bind preferentially to core sequence motifs with lengths of 8 nt (Puf3), 9 nt (Puf4), and 8–12 nt (Puf5) and shorter motifs correspond to increased curvature: Puf3 is the most extreme, Puf4 is intermediate, and Puf5 is relatively flat (*Gerber et al., 2004*; *Miller et al., 2008*; *Wilinski et al., 2015*; *Zhu et al., 2009*). The change in curvature of FBF-2 upon binding a compact RNA motif is especially unexpected given that yeast Puf5 maintained a fixed scaffold when bound to RNAs of 9–12 nt (*Wilinski et al., 2015*). The crystal structure of the FBF-2/LST-1 complex reveals a new means by which PUF protein RNA-binding specificity can be controlled: by changes in the curvature of the PUF protein. It is not clear whether LST-1 can direct this change in vivo, however, it and other protein-protein interactions have the potential to alter the topology of the PUF protein scaffold.

Previously, we generated mutations in the RNA-binding interface that direct the specificity of FBF-2 away from extended 9-nt elements and towards more compact 8-nt elements (*Bhat et al., 2019*). Importantly, there were no alterations in curvature. This implies that curvature is not required for a change in binding element length per se. The 8-nt sequences we used previously, bearing A4 and A5/U5, do not match the preferred compact element sequence we identified here and therefore did not induce FBF-2 curvature change. To test the notion that partner proteins preferentially associate with distinct conformational states, we attempted unsuccessfully to crystallize the binary FBF-2/LST-1 complex without RNA or assembled on an extended element. We were also unable to grow crystals of FBF-2 with the compact cFBE RNA. Additional experiments are required to definitively establish the effect(s) of protein partners on the conformation of the PUF scaffold.

Second, the FBF-2 curvature change fosters recognition of bases in the central region of the RNA-binding site. The ability to change curvature does not disrupt binding to high affinity sites but adds the ability to recognize compact elements with restricted sequence specificity for nucleotides in the central region of the motif. The 1:1 binding mode to the compact element includes recognition of nucleotides +4 and +5. Both G4 and A5 are recognized specifically in the FBF-2/LST-1/RNA crystal structure whereas there is more flexibility in recognition of nucleotides 4–6 in the extended FBF binding motif (*Wang et al., 2009*). Nucleotide A5 of the compact motif is bound by FBF-2 repeat 4 (*Figure 4A*). The RNA recognition side chains in FBF-2 repeat 4, NQ/H, would typically bind to a U5. However, A5 was preferred to U5 in our EMSAs, and in our FBF-2/LST-1/RNA crystal structure, the Hoogsteen edge of A5 is recognized by Q329. This is similar to the recognition of the Hoogsteen edge of an A4 in *COX17* site B RNA by *S. cerevisiae* Puf3, where A5 was also favored over U5 (*Zhu et al., 2009*). Therefore, FBF-2 and Puf3 utilize a common mechanism for higher affinity binding to a compact motif with a central A5 nucleotide.

Third, LST-1 binds to an interface on FBF-2 that appears to be utilized by two additional protein partners. A member of the cytoplasmic polyadenylation element binding protein family, CPB-1, controls spermatogenesis and interacts with FBF (*Luitjens et al., 2000*). Similarly, Germline Development Defective-3 (GLD-3) is a component of a cytoplasmic polyadenylation protein complex that also promotes spermatogenesis and binds to FBF (*Eckmann et al., 2004*). Both CPB-1 and GLD-3 require the loop between repeat 7 and repeat 8, including Y479, of FBF-2 (*Campbell et al., 2012b*; *Kim et al., 2009*; *Menichelli et al., 2013*). Intriguingly, the same residues mediate interactions between LST-1 and FBF-2.

Fourth and finally, LST-1 decreases the affinity of FBF for RNA. We envisage several potential implications on mRNA control. A general reduction in affinity would increase the importance of

sequence composition and mRNA expression level. Higher affinity or more abundant targets would be more likely to remain controlled in the presence of LST-1 while those with more degenerate sites or lower abundance would be lost. Therefore, LST-1 likely narrows the network of targets bound by FBF. Additionally, the restrictive pattern of LST-1 expression to the distalmost region of the germline provides a means to spatially restrict the FBF regulatory network within the stem cell region (*Shin et al., 2017*). Such a mechanism could enable remodeling of RNA-binding factors on mRNAs transported by FBF to sites of RNA processing. LST-1 is localized to perinuclear granules that are enriched for RNA nucleases and processing factors (*Shin et al., 2017*; *Smith et al., 2016*; *Wang et al., 2014*). We suggest that in addition to established roles for protein partners (*e.g.* Nanos) as clamps that enhance binding of PUF proteins to specific targets (*Weidmann et al., 2016*), partners can also facilitate target selection by acting as a rheostat through affinity reduction. This represents a new way that protein complexes can modulate the activity of RNA-binding scaffolds. Given their widespread occurrence in biology, it is perhaps unsurprising that protein partners can mediate multiple modes of allosteric transition that have potentially far-reaching impacts on RNA targeting and regulatory control.

# Materials and methods

**Key resources table**

| Reagent type (species) or resource | Designation | Source or reference | Identifiers | Additional information |
|---|---|---|---|---|
| Gene (*Caenorhabditis elegans*) | LST-1 | | UniprotKB: P91820_(CAEEL) | |
| Gene (*Caenorhabditis elegans*) | FBF-2 | | UniprotKB: Q09312_(CAEEL) | |
| Strain, strain background (*Saccharomyces cerevisiae*)) | L40 | ATCC | Cat. #: *MYA-3332* | Yeast 2-hybrid strain |
| Strain, strain background (*Escherichia coli*) | DH5-alpha | Thermo Fisher | Cat. #: 18265017 | Chemically competent cells |
| Strain, strain background (*Escherichia coli*) | BL21-CodonPlus (DE3)-RIL | Agilent | Cat. #: 230245 | Competent cells |
| Recombinant DNA reagent | pACT2 (plasmid) | PMID: 21372189 | GenBank Accession #: U29899 | Yeast two-hybrid expression vector with Gal4 activation domain fusion |
| Recombinant DNA reagent | pBTM116 (plasmid) | Clonetech | *Vojtek et al., 1993* | Yeast two hybrid vector with LexA DNA binding ORF |
| Recombinant DNA reagent | pSMT3 (plasmid) | provided by Dr. Christopher Lima | *Mossessova and Lima (2000)* | Encodes an N-terminal His$_6$-SUMO fusion tag followed by a TEV protease cleavage site |
| Recombinant DNA reagent | pGEX4T-3 (plasmid) | GE Healthcare | Cat. #: 27-4583-01 | Bacterial vector for expressing fusion proteins with a thrombin site |
| Recombinant DNA reagent | pMAL-C2T (plasmid) | New England Biolabs | Accession #: JF795283 | Bacterial vector for cytoplasmic expression of maltose-binding protein fusion |
| Sequence-based reagent | Yeast tRNA | Thermo Fisher | Cat. #: 15401011 | Carrier for nucleic acid precipitation |
| Peptide, recombinant protein | TURBO DNase | Thermo Fisher | Cat. #: AM2238 | |

*Continued on next page*

*Continued*

| Reagent type (species) or resource | Designation | Source or reference | Identifiers | Additional information |
|---|---|---|---|---|
| Peptide, recombinant protein | ImProm-II reverse transcription reaction | Promega | Cat. #: A3803 | |
| Peptide, recombinant protein | GoTaq reaction | Promega | Cat. #: M7123 | |
| Peptide, recombinant protein | T4 polynucleotide kinase | New England Biolabs | Cat. #: M0201S | |
| Peptide, recombinant protein | lysozyme | Thermo Fisher | Cat. #: 89833 | |
| Commercial assay or kit | β-Glo reagent | Promega | Cat. #: E4720 | |
| Commercial assay or kit | Phusion High-Fidelity PCR Kit | Thermo Fisher | Cat. #: F553S | |
| Commercial assay or kit | AmpliScribe T7-Flash Transcription Kit | Lucigen | Cat. #: ASF3507 | |
| Chemical compound, drug | EDTA-free Protease Inhibitor | Roche | Cat. #: 11836170001 | |
| Chemical compound, drug | Amylose resin | New England Biolabs | Cat. #: E8021S | |
| Chemical compound, drug | Glutathione agarose resin | Gold Biotechnology | Cat. #: G-250 | |
| Chemical compound, drug | Ni-NTA resin | Qiagen | Cat. #: 30210 | |
| Chemical compound, drug | reduced glutathione | Sigma-Aldrich | Cat. #: G4251 | |
| Chemical compound, drug | Glutathione magnetic beads | Thermo Fisher | Cat. #: 78602 | |
| Software, algorithm | HKL2000 | http://www.hkl-xray.com/ | *Otwinowski and Minor, 1997* | |
| Software, algorithm | Phaser | http://www.ccp4.ac.uk/html/phaser.html | *McCoy et al., 2007* | |
| Software, algorithm | Phenix | https://www.phenix-online.org | *Adams et al., 2010* | |
| Software, algorithm | Coot | https://www2.mrc-lmb.cam.ac.uk/personal/pemsley/coot | *Emsley and Cowtan, 2004* | |
| Software, algorithm | MEME | http://meme-suite.org/ | *Bailey et al., 2006* | |
| Software, algorithm | Enrichr | https://amp.pharm.mssm.edu/Enrichr/ | *Kuleshov et al., 2016* | |
| Software, algorithm | ImageQuant Version 5.1 | GE Healthcare | | |
| Software, algorithm | GraphPad Prism 7 | GraphPad | | |
| Software, algorithm | Matlab R2008a | MathWorks | | |

## Yeast molecular genetics

The RNA-binding region of FBF-2 corresponding to residues 121–632 was covalently fused to the GAL4 activation domain in the pACT2 vector (*Koh et al., 2011*). An identical construct was cloned into pBTM116 and used as a LexA DNA-binding domain fusion vector. LST-1 constructs were generated in the identical vectors and designated according to UniProt entry P91820. Our construct for LST-1 fragment 1–34 also encodes 70-aa residues upstream of the initiating methionine that were included in a previous annotation of the LST-1 open reading frame (UniProt entry P91820, version 109 and earlier). The yeast two-hybrid assays were performed in the L40 Ura- strain (*Bai and Elledge, 1997*; *Zhang et al., 1999*). Truncations and mutations were generated by site-directed mutagenesis (*Campbell and Baldwin, 2009*). Quantification of β-galactosidase activity was

accomplished using the β-Glo reagent (Promega) and detected using a 96-well Tecan Spark 20 plate reader. To account for differences in cell count, luminescence values were normalized to absorbance at 660 nm.

## Protein expression and purification

A cDNA fragment encoding the PUM domain of *C. elegans* FBF-2 (residues 164–575) was cloned into the pSMT3 vector (kindly provided by Dr. Christopher Lima), which encodes an N-terminal His$_6$-SUMO fusion tag followed by a TEV protease cleavage site (*Mossessova and Lima, 2000*). A cDNA fragment encoding amino acid residues 74–98 of LST-1 was PCR-amplified and cloned into the pGEX4T-3 vector with a TEV site after the glutathione *S*-transferase (GST)-tag. The two recombinant plasmids were co-transformed into BL21-CodonPlus (DE3)-RIL competent cells (Agilent) using both kanamycin and ampicillin for selection. A 5 ml culture was grown from colonies overnight at 37°C and then used to inoculate 1 l LB media with 50 µg/ml kanamycin and 100 µg/ml ampicillin at 37°C. Protein expression was induced at OD$_{600}$ of ~0.6 with 0.4 mM IPTG at 16°C for 16–20 hr.

The cell pellet was resuspended in 40 ml lysis buffer containing 20 mM Tris, pH 8.0; 0.5 M NaCl; 20 mM imidazole; 5% (v/v) glycerol; and 0.1% (v/v) β-mercaptoethanol and disrupted by sonication. After centrifugation the soluble lysate was mixed with 5 ml Ni-NTA resin (Qiagen) in a 50 ml conical tube rotating at 4°C for 1 hr. The mixture was then transferred into a Bio-Rad Econo-Pac gravity column. The beads were washed with 300 ml lysis buffer. The His$_6$-SUMO-FBF-2 and GST-LST-1 fusion proteins were co-eluted with ~70 ml elution buffer (20 mM Tris, pH 8; 50 mM NaCl; 200 mM imidazole, pH 8; 1 mM dithiothreitol [DTT]). TEV protease was added to the eluent and incubated at 4°C overnight. The His$_6$-SUMO fusion was cleaved from FBF-2 and the GST fusion was cleaved from LST-1. Subsequently, the protein solution was filtered through a 0.22 µM filter and loaded onto a 5 ml Hi-Trap Heparin column (GE Healthcare). Heparin column buffer A contained 20 mM Tris, pH 8 and 1 mM DTT, and buffer B contained an additional 1 M NaCl. The proteins were eluted with a salt gradient of 5–100% buffer B. The fractions containing both FBF-2 and LST-1 (eluted at about 32% buffer B) were pooled and concentrated using Amicon protein concentrators with a 30 kDa molecular weight cutoff. 500 µl concentrated protein complex was loaded onto a Superdex 75 10/300 GL column (GE Healthcare) equilibrated in 20 mM HEPES, pH 7.4; 0.15 M NaCl; and 2 mM DTT. The protein complex of FBF-2 and LST-1 eluted at a volume of 11.2 ml. The proteins were concentrated to OD$_{280}$ of ~4.0.

For EMSAs, FBF-2 protein was purified as described previously (*Bhat et al., 2019*). GST-tagged LST-1 protein (residues 67–98) was overexpressed in *E. coli* BL21-CodonPlus (DE3)-RIL cells by induction with 0.4 mM IPTG at 37°C for 3 hr. The cell pellet was resuspended in PBS buffer and disrupted by sonication. After centrifugation the soluble lysate was mixed with 2 ml GST resin for 1 hr at 4°C. The resin was washed with PBS buffer before the protein was eluted with 50 mM Tris (pH 8.0), 50 mM NaCl, 10 mM reduced glutathione, and 1 mM DTT. TEV protease was added to the eluent and incubated overnight. LST-1 was separated from cleaved GST by a heparin column and further purified by a HiLoad 16/60 Superdex 75 column in the buffer of 20 mM HEPES, pH 7.4, 150 mM NaCl, 2 mM DTT. The protein was concentrated to 400 µM for EMSA.

## Crystallization

The concentrated protein complex was mixed with RNA (5′-CUGUGAAU-3′) at a molar ratio of 1:1.2 and incubated on ice for 1 hr prior to crystallization screening. Crystals of the ternary complex of FBF-2/LST-1/RNA were obtained in the condition of 17–20% (w/v) PEG 3350, 0.2 M MgCl$_2$, 0.1 M MES, pH 6.5 by hanging drop vapor diffusion at 20°C with a 1:1 ratio of sample:reservoir solution. Crystals were cryoprotected by transferring them into a series of the crystallization solution supplemented with 5%, 10%, or 20% (v/v) ethylene glycol and flash freezing them in liquid nitrogen.

## Data collection and structure determination

X-ray diffraction data were collected at beamline 22-ID of the Advanced Photon Source. Data sets collected from two similar crystals at the wavelength of 1.0 Å were scaled together with HKL2000 (*Otwinowski and Minor, 1997*) to improve the data completeness. The crystals belong to the P1 space group. The structure of FBF-2 with 5′-UGUG (modified from PDB code: 3v74) was used as a search model for molecular replacement with Phaser (*McCoy et al., 2007*). The model was improved

through iterative refinement and building with Phenix and Coot (*Adams et al., 2010*; *Emsley and Cowtan, 2004*). The LST-1 peptide was then built into the density and further refined to final $R_{work}$/$R_{free}$ of 0.198/0.240 at 2.1 Å resolution. An asymmetric unit contains two sets of ternary complexes. Complex A contains protein residues 167–375, 382–568 of FBF-2, 76–90 of LST-1 and RNA nucleotides 1–8. Complex B contains protein residues 167–375, 382–523, 528–564 of FBF-2, 75–90 of LST-1 and RNA nucleotides 2–8. Data collection and refinement statistics are shown in *Table 1*.

## SEQRS and bioinformatics

SEQRS was used to analyze the specificity of the FBF-2/LST-1 complex as described with minor adjustments (*Campbell et al., 2012a*; *Lou et al., 2017*; *Zhou et al., 2018*). FBF-2 (residues 163–632) was cloned into pMAL-C2T vector, resulting in an N-terminal fusion to the maltose-binding protein (MBP). LST-1 (residues 34–180) was cloned into modified pGEX4T-1 to express recombinant LST-1 with N-terminal GST and C-terminal His$_6$ tags. Recombinant proteins were expressed and purified separately using a similar approach. Briefly, bacterial cells were pelleted and resuspended in lysis buffer (50 mM Tris-HCl, pH 8.0; 500 mM NaCl; 5 mM DTT; 1 mM EDTA; 5% [v/v] glycerol; and 0.1% [v/v] NP-40) with the addition of 1 mg/ml lysozyme, 0.17 mg/ml PMSF, and Complete EDTA-free Protease Inhibitor (Roche). The supernatant fractions obtained from centrifugation were incubated with amylose resin (New England BioLabs) for FBF-2 recombinant protein or glutathione agarose resin (Gold Biotechnology) for LST-1 recombinant protein, respectively, at 4°C, followed by three washes with lysis buffer. FBF-2 was recovered with the application of an elution buffer (50 mM Tris-HCl, pH 8.0; 300 mM NaCl; 5 mM DTT; 5% [v/v] glycerol; 30 mM reduced-glutathione). LST-1 was eluted using the same buffer but 10 mM maltose replaced the reduced-glutathione. Purified proteins were dialyzed in a buffer containing 10 mM Tris-HCl, pH 8.0; 300 mM NaCl; 5% (v/v) glycerol; and 17 µg/ml PMSF for 16 hr at 4°C and concentrated with a 30,000 Da cutoff Amicon Ultra-15 centrifugal filter (Sigma-Aldrich). Complexes were generated through capture of FBF-2 by 2 nmol LST-1 immobilized on glutathione magnetic beads (Thermo Fisher Scientific). The initial dsDNA pool containing random sequence of 20-nt was generated by PCR amplification using the primer set and degenerate DNA oligo (IDT) template described previously (*Lou et al., 2017*). The RNA library was obtained following transcription of 0.5 µg of dsDNA using the AmpliScribe T7-Flash Transcription Kit (Lucigen). DNA was removed from the initial library through the addition of 2 units of TURBO DNase (Thermo Fisher Scientific) and incubation for 1 hr at 37°C. An 800 ng aliquot of the resulting library was allowed to bind to the FBF-2/LST-1 complex in SEQRS buffer (25 mM Tris-HCl, pH 8.0; 1 mM EDTA; 150 mM NaCl; 5 mM DTT; 1% [v/v] glycerol and 0.01% [v/v] NP-40) for 30 min at 22°C in the presence of 200 ng yeast tRNA (Thermo Fisher Scientific). Bound RNAs were enriched by washing the complex with 200 µl of ice cold SEQRS buffer for four iterations. After the wash steps were complete, beads were incubated with 20 µl elution buffer (1 mM Tris-HCl, pH 7.5) that contained 10 pmol reverse transcription primer for 10 min at 65°C and ice chilled for 2 min. 5 µl of the elution was added to a separate tube containing 10 µl of an ImProm-II reverse transcription reaction (Promega). After incubation for 60 min at 42°C, the cDNA was used as a template for PCR in a 50 µl GoTaq reaction (Promega). The SEQRS cycle was repeated five times, and the Illumina flow cell adapter sequence was added to dsDNA in the final PCR amplification. Sequencing was conducted at the UT-Dallas Genome Center, and the data were analyzed as described (*Weidmann et al., 2016*). Sequence logos were generated using MEME (*Bailey et al., 2006*). Bioinformatics on CLIP and SEQRS data was done using pattern matching with the grep PERL function in command line. The compact and extended patterns were defined as CTGTRNAT or CTGTRNNAT, respectively. To identify sites of FBF association to mRNAs containing the compact or extended motifs, we examined FBF-1 and FBF-2 iCLIP peaks (*Prasad et al., 2016*). Genes containing peaks with exact matches were collated into gene sets and were analyzed for functional relatedness using Enrichr (*Kuleshov et al., 2016*) with the Benjamini-Hochberg method for P-value correction (*Kuleshov et al., 2016*).

## Electrophoretic mobility shift assays

Synthetic RNAs (GE Dharmacon) were labeled with $^{32}$P- γ-ATP by T4 polynucleotide kinase for 1 hr at 37°C. Unincorporated $^{32}$P- γ-ATP was removed using Illustra MicroSpin G-25 columns. Radiolabeled RNAs (100 pM) were mixed with serially diluted protein samples in 10 mM HEPES (pH 7.4), 50

mM NaCl, 0.01% (v/v) Tween-20, 0.1 mg/ml BSA, 0.1 mg/ml yeast tRNA and 2 mM DTT. The FBF-2 protein concentrations were: 4000, 2000, 1000, 500, 250, 125, 62.5, 31.2, 15.6, 7.8, 3.9, 1.95, 0.98, 0.49, 0 nM. In parallel, purified LST-1 protein (10X stock concentration) was added to the FBF-2 protein series to a final concentration of 4 μM throughout, and the FBF-2 concentrations were adjusted by multiplying by factor of 0.9 during data analysis. Binding reactions were incubated at 4°C overnight. The samples were resolved on 10% TBE polyacrylamide gels run at constant voltage (100 V) with 1X TBE buffer at 4°C for 35 min. The gels were dried and visualized using a Typhoon Phosphor-Imager (GE Healthcare). Band intensities were quantified with ImageQuant 5.1. The data were fit with GraphPad Prism 7 using nonlinear regression with a one-site specific binding model. Mean $K_d$'s and standard error of the mean from three technical replicates are reported (*Table 2*).

## Acknowledgements

We thank John Gonczy for assistance with data collection at SER-CAT beamlines 22-ID and 22-BM at the Advanced Photon Source, Argonne National Laboratory, Lars Pedersen and Juno Krahn for crystallographic and data collection support at NIEHS, and Jason Williams and the staff of the NIEHS Mass Spectrometry Research and Support Group for mass spectrometric analyses. We appreciate critical reading of this manuscript by our colleagues K McCann and R Stanley. This work was supported in part by NIH grant R01NS100788 (ZTC) and by the Intramural Research Program of the National Institutes of Health, National Institute of Environmental Health Sciences (TMTH). The Advanced Photon Source used for this study was supported by the US Department of Energy, Office of Science, Office of Basic Energy Sciences, under contract no. W-31–109-Eng-38.

## Additional information

### Funding

| Funder | Grant reference number | Author |
| --- | --- | --- |
| National Institutes of Health | ZIA ES50165 | Traci M Tanaka Hall |
| National Institutes of Health | R01NS100788 | Zachary T Campbell |

The funders had no role in study design, data collection and interpretation, or the decision to submit the work for publication.

### Author contributions

Chen Qiu, Vandita D Bhat, Zachary T Campbell, Traci M Tanaka Hall, Conceptualization, Supervision, Funding acquisition, Investigation, Writing—original draft, Project administration, Writing—review and editing; Sanjana Rajeev, Conceptualization, Supervision, Investigation, Writing—original draft, Writing—review and editing; Chi Zhang, Investigation, Writing—original draft, Writing—review and editing; Alexa E Lasley, Robert N Wine, Investigation, Writing—review and editing

### Author ORCIDs

Zachary T Campbell https://orcid.org/0000-0002-3768-6996
Traci M Tanaka Hall https://orcid.org/0000-0001-6166-3009

### Decision letter and Author response

Decision letter https://doi.org/10.7554/eLife.48968.032
Author response https://doi.org/10.7554/eLife.48968.033

## Additional files

### Supplementary files

• Transparent reporting form
DOI: https://doi.org/10.7554/eLife.48968.025

## Data availability

Atomic coordinates and structure factors are deposited under RCSB PDB ID 6PUN. SEQRS sequence data are available through the Dryad Digital Repository, https://doi.org/10.5061/dryad.30501q7.

The following datasets were generated:

| Author(s) | Year | Dataset title | Dataset URL | Database and Identifier |
|---|---|---|---|---|
| Qiu C, Bhat VD, Rajeev S, Zhang C, Lasley AE, Wine RN, Campbell ZT, Hall TMT | 2019 | SEQRS data for FBF-2 and SEQRS data for the LST-1 FBF-2 complex | https://doi.org/10.5061/dryad.30501q7 | Dryad Digital Repository, 10.5061/dryad.30501q7 |
| Qiu C, Bhat VD, Rajeev S, Zhang C, Lasley AE, Wine RN, Campbell ZT, Hall TMT | 2019 | Crystal structure of a ternary complex of FBF-2 with LST-1 (site B) and compact FBE RNA | https://www.rcsb.org/structure/6PUN | Protein Data Bank, 6PUN |

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
