## [Decision Letter]

Thank you for submitting your article "A crystal structure of a collaborative RNA regulatory complex reveals mechanisms to refine target specificity" for consideration by *eLife*. Your article has been reviewed by James Manley as the Senior Editor, Timothy Nilsen as the Reviewing Editor, and two reviewers. The reviewers have opted to remain anonymous.

The reviewers have discussed the reviews with one another and the Reviewing Editor has drafted this decision to help you prepare a revised submission.

Both reviewers found the work to be interesting and well done. Nevertheless, both made several suggestions for improvement. Please address these points as thoroughly as possible. Reviews are below.

*Reviewer #1:*

This is an excellent manuscript by Campbell, Hall, and colleagues that demonstrates how binding of a co-factor protein (LST-1) to an RNA-binding protein (FBF-2) can alter its apparent binding specificity. In this case, LST-1 binds as an extended peptide to the interface between repeat 7 and repeat 8. This binding alters the curvature of FBF-2, which has profound impacts on its sequence preferences and overall affinity Data includes yeast two hybrid interaction mapping, high resolution crystallography, SEQRS to map binding specificity, and quantitative EMSA studies. The data paint a compelling picture that binding alters affinity and specificity. Mutagenesis studies show that some key features of the interface are critical for the interaction, while others are dispensable. The authors predict that other FBF-2 interaction partners use a similar interface based on the presence of similar amino acid sequences. All told, I find the manuscript to be clear, convincing, and worthy of publication in *eLife*.

I have a few minor suggestions for the authors to consider:

1) It appears FBF-1 binds more weakly to LST-1 than FBF-2 in the Y2H data. Can this difference be rationalized in terms of differences in protein sequence?

2) A summary of the published data that supports the interaction between CPB-2 and GLD-3 with the LST-1 binding site in FBF-2 would be useful.

3) I would have preferred to see more introduction to the lst-1 phenotype and how it ties into the fbf-1 phenotype. What is known about how critical lst-1 is to worm reproduction?

4) What is the affinity of the interaction between LST-1 and FBF-2? Is it tight enough that binding will occur as an obligate heterodimer, or is there evidence of cooperativity? Are there regions outside of the interacting domain that could potentially interact with RNA non-specifically (patches of lysine/arginine, etc). If so, are these regions present in the binding affinity studies?

*Reviewer #2:*

The manuscript by Qiu and co-workers reports on the tuning of the interaction between the C.elegans RNA binding protein FBF and the RNA targets by the regulator LST-1. The authors define the minimal interacting region on LST-1 and crystalize a ternary complex comprising this region, the PUF domain of FBF and a cognate RNA sequence. Based on the structure and on data on similar proteins they design a number of mutations to define the key contact in the interaction. Then, they examine RNA binding, using again structural information and a SELEX-type assay and they report that binding of the regulatory peptide changes the curvature of the RNA-binding surface as well as the recognition of the nucleotide at the 3' of the sequence, leading to a shorter consensus sequence. The paper is connected to a previous study by the authors where FBF-RNA binding features are manipulated using structure-driven mutant design.

The manuscript presented here is streamlined and overall well written. The Figures are clear. The data, as far as I can judge are of high quality. The paper had several points of interest, including the description of the interface and key contacts between FBF and its regulator and the effect on RNA binding. How mechanistically the regulator constrains FBF to change its curvature upon RNA binding, whether this effect is visible in the absence of RNA and what is the functional significance of this effect is to be established. I have a few queries:

1) Subsection “LST-1 contacts the non-RNA binding surface of FBF-2 via conserved interaction hot Spots” – the authors state that mutation of K80 to alanine (hotspot 2) has essentially no effect on binding. Shortly after, they discuss that 'in contrast to hotspots 1 and 2 hotspot 3 was largely dispensable for binding'. I find these two statements are contradictory. My understanding of the data is that mutation of amino acids 83 and 85 results in a very significant effect while mutation of the other tested amino acids does not, and that the region of the peptide amino-terminal to 80 less important for recognition. This is consistent with the data on the deletion constructs (constructs 35-80 and 80-180), which should also be discussed by the authors in this context.

2) Based on the SELEX data, the protein recognises a shorter sequence in the presence of the regulator. This is consistent with the lower affinity for the RNA reported for the complex. Further, based on the shorter length of the recognized sequence sequence and on the smaller enrichment ratios reported in Table 3 I would say the specificity also decreases. The authors could discuss this.

3) The analysis of the iCLIP data represent the connection with the recognition of the RNA target in the cell and is important to define the physiological relevance of the observation made, particularly considering that many of the difference reported are not large. This analysis and the GO analysis should be more detailed as at present is difficult to evaluate the authors conclusion.

---

## [Author Response]

Reviewer #1:

[…] I have a few minor suggestions for the authors to consider:1) It appears FBF-1 binds more weakly to LST-1 than FBF-2 in the Y2H data. Can this difference be rationalized in terms of differences in protein sequence?

We have found that FBF-1 is consistently expressed at lower levels than FBF-2. For this reason, FBF-2 is more tractable in heterologous binding measurements in yeast given the superior signal-to-noise ratio relative to FBF-1. However, there are differences in protein sequence. In comparing the region from R441 to F552, there are only 6 substitutions. L465M is a buried hydrophobic core residue, I480M is exposed and near LST-1 I86 but the substitution is very conservative, D484G is in the R7-R8 loop not near the peptide, G524S is in the loop between helices 2 and 3 of repeat 8 and not near LST-1, I529N and D533V are in helix 3 of repeat 8 and exposed and not near LST-1. Possibly a longer LST-1 could interact with this part of repeat 8, but it is quite far away from the minimal binding site that we see. A definitive answer would require more precise in vitro binding measurements, which we would like to pursue in future work.

2) A summary of the published data that supports the interaction between CPB-2 and GLD-3 with the LST-1 binding site in FBF-2 would be useful.

This is an excellent point, and we have added additional details that highlight similarities between CPB-1 and GLD-3 and to the Discussion.

3) I would have preferred to see more introduction to the lst-1 phenotype and how it ties into the fbf-1 phenotype. What is known about how critical lst-1 is to worm reproduction?

Individually, loss of either LST-1 or FBF-1 results in minor to no detectable phenotype. However, both genes are critical when depleted in sensitized backgrounds, either SYGL-1 or FBF-2, respectively. The phenotypes are related as they converge on loss of GSC renewal. This similarity is described in the third paragraph of the Introduction.

4) What is the affinity of the interaction between LST-1 and FBF-2? Is it tight enough that binding will occur as an obligate heterodimer, or is there evidence of cooperativity? Are there regions outside of the interacting domain that could potentially interact with RNA non-specifically (patches of lysine/arginine, etc). If so, are these regions present in the binding affinity studies?

In response to this comment, we performed a preliminary measurement of the LST-1 residues 72-90/FBF-2 affinity by fluorescence polarization and determined that it is ~2.7 μM. This is similar to what was measured for peptides from CPB-1 and GLD-3. We are planning a future study to examine the partner protein interactions with FBF-2, and also FBF-1 if we successfully express the protein. We aim to test different length fragments, including LST-1 1-152, which Judith Kimble’s lab showed to be sufficient for germline stem cell maintenance. We hope to be able to express this longer protein for future studies. There are basic regions in LST-1 near the motifs that interact with FBF-2. The peptide used in EMSAs does contain some of the patches (LST-1 67-98 (basic residues in bold and interaction motif in italics, SSSPQQ**R**SGL**R**SQ*KLHL*TYIE**K**N**KR**V**R**AMIPQ). We agree that it is a good idea to test in the future the importance of these basic patches for RNA interaction. Thank you for the suggestion.

Reviewer #2:

[…] The manuscript presented here is streamlined and overall well written. The Figures are clear. The data, as far as I can judge are of high quality. The paper had several points of interest, including the description of the interface and key contacts between FBF and its regulator and the effect on RNA binding. How mechanistically the regulator constrains FBF to change its curvature upon RNA binding, whether this effect is visible in the absence of RNA and what is the functional significance of this effect is to be established. I have a few queries:1) Subsection “LST-1 contacts the non-RNA binding surface of FBF-2 via conserved interaction hot Spots” – the authors state that mutation of K80 to alanine (hotspot 2) has essentially no effect on binding. Shortly after, they discuss that 'in contrast to hotspots 1 and 2 hotspot 3 was largely dispensable for binding'. I find these two statements are contradictory. My understanding of the data is that mutation of amino acids 83 and 85 results in a very significant effect while mutation of the other tested amino acids does not, and that the region of the peptide amino-terminal to 80 less important for recognition. This is consistent with the data on the deletion constructs (constructs 35-80 and 80-180), which should also be discussed by the authors in this context.

We apologize that this section was confusing due to our mislabeling of the figure and a typo in the text. We hope that the corrections clarify our findings and no longer give the impression of contradictory statements. Hotspot 2 includes two interactions. The interaction including K80 is not important (K80A has no effect on FBF-2/LST-1 binding), but the second interaction including Q448 is somewhat important (Q448G reduces FBF-2/LST-1 binding). We agree with the reviewer’s conclusions about the regions of highest importance for binding. We now explicitly state the relative importance of residues C-terminal to K80 and connect to the deletion construct data.

2) Based on the SELEX data, the protein recognises a shorter sequence in the presence of the regulator. This is consistent with the lower affinity for the RNA reported for the complex. Further, based on the shorter length of the recognized sequence sequence and on the smaller enrichment ratios reported in Table 3 I would say the specificity also decreases. The authors could discuss this.

We have added a sentence to subsection “LST-1 imparts distinct RNA-binding selectivity to FBF-2” and edited the Discussion section to clarify our interpretation of the SEQRS results. Specifically, we explain that although the motif appears shorter, the recognized sequence in the crystal structure includes recognition of the 3´ sequence that is not part of the conserved motif. In addition, the EMSA data indicate that affinities are sensitive to changes in the 3´ end (cFBE-7 versus cFBE and cFBE G8A). For Table 3, it is important to take into consideration that LST-1 is not required for interaction with the cFBE motif. The weakened affinity makes it difficult to place too much emphasis on the FBF-2/LST-1 SEQRS results. Because we detected no influence on sequence specificity due to LST-1, the SEQRS data for FBF-2 alone can be used to make conclusions about sequence preferences in the FBE and cFBE motifs.

3) The analysis of the iCLIP data represent the connection with the recognition of the RNA target in the cell and is important to define the physiological relevance of the observation made, particularly considering that many of the difference reported are not large. This analysis and the GO analysis should be more detailed as at present is difficult to evaluate the authors conclusion.

We have now provided additional details on the genes identified in the CLIP-seq experiments and the GO analysis in the Materials and methods section and by including the corresponding gene sets in the source data file.